# Correlation between Pulmonary Artery Pressure and Vortex Duration Determined by 4D Flow MRI in Main Pulmonary Artery in Patients with Suspicion of Chronic Thromboembolic Pulmonary Hypertension (CTEPH)

**DOI:** 10.3390/jcm11175237

**Published:** 2022-09-05

**Authors:** Jean-François Deux, Lindsey A. Crowe, Léon Genecand, Anne-Lise Hachulla, Carl G. Glessgen, Stéphane Noble, Maurice Beghetti, Jin Ning, Daniel Giese, Frédéric Lador, Jean-Paul Vallée

**Affiliations:** 1Division of Radiology, Diagnostic Department Geneva University Hospitals, 1205 Geneva, Switzerland; 2Faculty of Medicine, University of Geneva, 1205 Geneva, Switzerland; 3Pulmonary Hypertension Program, Geneva University Hospitals, 1205 Geneva, Switzerland; 4Division of Pulmonary Medicine, Department of Medicine, Geneva University Hospitals, 1205 Geneva, Switzerland; 5Division of Cardiology, Department of Medicine, Geneva University Hospitals, 1205 Geneva, Switzerland; 6Paediatric Cardiology Unit, Geneva University Hospitals, 1205 Geneva, Switzerland; 7Centre Universitaire Romand de Cardiologie et Chirurgie Cardiaque Pédiatrique, University of Geneva and Lausanne, 1205 Geneva, Switzerland; 8Siemens Medical Solutions USA Inc., Cleveland, OH 44125, USA; 9Magnetic Resonance, Siemens Healthcare GmbH, 91052 Erlangen, Germany

**Keywords:** cardiac MRI, 4D flow MRI, pulmonary hypertension, vortical flow, vortex duration, chronic thromboembolic disease, right heart catheterisation

## Abstract

Chronic thromboembolic pulmonary hypertension (CTEPH) is one of the causes of pulmonary hypertension (PH) and requires invasive measurement of the mean pulmonary artery pressure (mPAP) during right heart catheterisation (RHC) for the diagnosis. 4D flow MRI could provide non-invasive parameters to estimate the mPAP. Twenty-five patients with suspected CTEPH underwent cardiac MRI. Mean vortex duration (%), pulmonary distensibility, right ventricular volumes and function were measured using 4D flow MRI and cine sequences, and compared with the mPAP measured by RHC. The mPAP measured during RHC was 33 ± 16 mmHg (10–66 mmHg). PH (defined as mPAP > 20 mmHg) was present in 19 of 25 patients (76%). A vortical flow was observed in all but two patients (92%) on 4D flow images, and vortex duration showed good correlation with the mPAP (r = 0.805; *p* < 0.0001). Youden index analysis showed that a vortex duration of 8.6% of the cardiac cycle provided a 95% sensitivity and an 83% specificity to detect PH. Reliability for the measurement of vortex duration was excellent for both intra-observer ICC = 0.823 and inter-observer ICC = 0.788. Vortex duration could be a useful parameter to non-invasively estimate mPAP in patients with suspected CTEPH.

## 1. Introduction

Pulmonary hypertension (PH) is a haemodynamic condition defined as a resting mean pulmonary artery pressure (mPAP) above 20 mmHg, as measured by right heart catheterisation (RHC) which is the gold standard for pressure measurement [1]. PH can arise from multiple conditions, including pulmonary and left heart diseases. More rarely, it may be due to a proliferative disease involving pulmonary vessels (pulmonary arterial hypertension, PAH) or to chronic thromboembolic pulmonary hypertension (CTEPH) [2]. CTEPH can lead to right ventricular failure and is associated with a poor prognosis if not treated [3]. Therefore, the early detection of this pathology is a major challenge in order to set up an early treatment and improve patient survival, in particular before right ventricular dysfunction occurs [4]. Non-invasive imaging may play an important role to meet this challenge [5]. Transthoracic echocardiography is currently the most commonly used screening technique to determine PH probability [6]. RHC remains the gold standard for measuring the mean pulmonary artery pressure (mPAP). Other non-invasive techniques as alternatives to echocardiography could be useful for the diagnostic work up of PH. Several authors have already assessed the ability of cardiac MRI to detect PH and to quantify it, based on the right ventricular function and mass, the curvature of the interventricular septum [7], the flow rate across the pulmonary artery [8], the wall shear stress [9,10] and the presence of vortices in the pulmonary artery using 4D flow MRI [11,12,13]. In addition, it has been reported that the duration of the vortices in the pulmonary artery was related to the mPAP and could be used to detect and quantify PH [14,15,16,17,18], in patients with different causes of PH.

In this study, we hypothesised that abnormalities of duration of vortical flow in the main pulmonary artery should be encountered on 4D flow MRI in a specific population of patients with suspected CTEPH, and could be a marker of PH. We therefore evaluated the changes in pulmonary flow on MRI in a population of patients with suspected CTEPH, and compared our results with other MRI parameters as well as with the values measured by RHC.

## 2. Materials and Methods

### 2.1. Study Population

All consecutive patients with suspected CTEPH and evaluated by both cardiac MRI and RHC at our institution over a period of 3 years were considered for inclusion in this retrospective study. The suspicion of CTEPH was based on the presence of suggestive symptoms and pulmonary perfusion anomalies as assessed by at least ventilation/perfusion scintigraphy and CT angiography as recommended [19]. Criteria of exclusion were: no patient agreement for the use of the data (*n* = 5), incomplete MR investigation (*n* = 8) and a delay > 6 months between cardiac MRI and RHC (*n* = 2). In total, 25 patients were included (Figure 1).

### 2.2. Cardiac MRI

All studies took place using a 3T system (PrismaFit; Siemens Healthcare; Germany). The imaging protocol included cine MR balanced Steady State Free Precession (bSSFP) sequences acquired in the long axis planes of the right ventricle (1 horizontal long axis (HLA) and 1 vertical long axis (VLA)) and contiguous sections acquired in the short axis plane for evaluation of right ventricular function. Following parameters were used for the acquisition: temporal resolution = 20 or 40 ms according to the cardiac rhythm of the patient; TE = 1.5 ms; pixel size = 2 mm × 2 mm × 8 mm, 25 reconstructed cardiac phases, flip angle = 30°. A 2D phase contrast flow sequence was acquired on the pulmonary artery (single slice) with the following parameters: temporal resolution = 28 ms; TE = 2.5 ms; pixel size = 2 × 2 × 6 mm; flip angle = 20°; velocity encoding (venc) = 150 cm/s. Lastly, a prototype 4D phase contrast flow sequence (Work In Progress number: 785) was acquired in the oblique sagittal orientation after contrast medium injection (0.2 mmol/kg of Dotarem; Guerbet; France) with the following parameters: prospective cardiac gating; 60 slices, temporal resolution of 20–40 ms (depending on patient physiology), 24 ± 5 cardiac phases, pixel size = 2 × 2 × 2 mm, 3 directions of flow quantification, venc = 150 cm/s. Flow images were acquired during free breathing using a liver dome respiratory navigator; mean duration of the sequence was 16 ± 7 min.

### 2.3. Right Heart Catheterisation (RHC)

The RHC exam was performed in a supine position with continuous monitoring of the electrocardiogram and arterial oxygen saturation using pulse oximetry (SpO2). The modified Seldinger technique was used for venous catheterisation of the femoral (7F Terumo introducer, Tokyo, Japan), basilic or cephalic vein (7F Terumo Glidesheath Slender radial introducer sheath, which has an outer diameter of a 6F sheath.) The Swan Ganz catheter (7F) allowed the resting haemodynamic evaluation with measurements of mPAP, pulmonary capillary wedge pressure (PCWP) and cardiac output (CO) as determined by thermodilution (TD). Left heart catheterisation with measurement of left ventricular end diastolic pressure (LVEDP), systemic arterial pressure was also routinely performed. The coronary arteries were injected only if clinically indicated. The mid-thoracic line was used for the zero-level reference. Pulmonary vascular resistance (PVR) and cardiac index (CI) were calculated with the respective formulae: PVR = (mPAP − PCWP)/CO or (mPAP − LVEDP)/CO if LVEDP was measured; CI = cardiac output/body surface area. TD was performed with 10 mL of iced, cold, sterile isotonic glucosaline solution injected in the proximal catheter’s lumen. The temperature change was recorded at the distal end of the probe with a thermistor. Measurements were performed in triplicate, and the mean value was recorded if the difference between the highest and lowest value was ≤10%. Otherwise, 2 more measurements were performed after exclusion of the highest and lowest values. The mean of those three remaining values was then calculated.

### 2.4. Image Analysis

Cine sequences were analysed with a dedicated software (CVI; circle) and following MRI parameters were recorded: right ventricular end diastolic volume (RV EDV; mL/m^2^), right ventricular end systolic volume (RV ESV; mL/m^2^), right ventricular ejection fraction (RVEF; %) and right ventricular cardiac index (RV cardiac index; L/min/m^2^).

The mean diameter of the pulmonary artery (mm) as calculated from maximal and minimum diameters on diastolic phase was recorded. Diastolic and systolic pulmonary artery cross-section area (mm^2^) was measured on the 2D flow sequence, the pulmonary artery distensibility, defined as the variation of surface between diastolic and systolic phase (%) was calculated, and the peak velocity in the pulmonary artery.

For 4D flow image analysis, streamlines in the pulmonary artery were automatically reconstructed using a dedicated prototype software (4D flow, Siemens Healthcare; Germany). A background phase correction was used to reconstruct the images. One operator (LC), blinded to clinical information, performed a visual analysis of all images. They considered that the vortex was present in the pulmonary artery with the appearance of a circular formation, whose axis of rotation is perpendicular to the axis of the vessel. The end of the vortex was defined as the last image showing this structure in the artery. The presence of a vortical blood flow was noted and its relative duration with respect to the whole cardiac cycle (%) was calculated. A vortical blood flow was defined as a closed concentric ring-shaped flow with the vortical blood flow’s axis of rotation being perpendicular to the pulmonary artery (PA) on visual analysis, as previously reported [16]. A second visual analysis of 4D flow images was performed 3 months later in a blinded fashion by the same operator in order to assess the intra-observer reproducibility. A second operator (ALH) performed a visual analysis of 4D flow images of all patients in a blinded fashion in order to calculate inter-observer reproducibility.

### 2.5. Statistical Analysis

Data are presented as mean ± SD, or median (interquartile range) depending on data distribution, or percentage. Differences between continuous data were tested with the Mann–Whitney rank-sum test for two-group comparisons. Pearson correlation coefficients (r) were calculated to assess the correlation between continuous variables. Simple regression analysis test was performed between mPAP measurements by RHC and MRI-derived vortex duration for patients with a mPAP > 20 mm Hg. Comparison between mPAP values measured by RHC and those calculated from the vortex duration by the regression equation was performed by Bland–Altman analysis. A multiple regression analysis was performed between mPAP values, vortex duration and MRI parameters that were non-significantly correlated with vortex duration. Receiver operating characteristic (ROC) analysis was applied to MR calculated parameters to assess the performance of each parameter to detect patient with PH. In order to facilitate comparison with studies that used a 25 mmHg cut-off to define PH, two different cut-offs were tested to define PH (>20 mmHg and ≥25 mmHg). The DeLong test was carried out to compare the ROC curves. A Youden index was used to evaluate the sensibility and specificity of different thresholds of vortex duration for the diagnosis of PH. Inter- and intra-observer intraclass correlation coefficient (ICC) were calculated for all subjects to evaluate the reproducibility of measurement of vortex duration [20]. A *p* value < 0.05 was considered as significant. Analyses were performed with the SPSS software (version 20.0, IBM SPSS Statistics; Chicago, IL, USA).

## 3. Results

### 3.1. Population Characteristics

Twenty-five patients with suspected CTEPH were included. The mean age was 63 ± 16 years and 10 patients were male (42%). Patient characteristics, including the results of RHC, are reported in Table 1. Mean mPAP was 33 ± 16 mmHg (10–66 mmHg) on RHC. PH was present in 19 of 25 patients (76%) for a threshold of 20 mmHg, as recently recommended [1], and in 16 patients (64%) if using a threshold of pulmonary pressure ≥ 25 mmHg [19]. A final diagnosis of CTEPH was retained in 19 patients. Among the remaining patients, 2 had a chronic thromboembolic disease without pulmonary hypertension, 2 had primary arterial hypertension (group 1), 1 had a PH related to left heart disease (group 2) and 1 patient had no PH and no definite diagnosis.

### 3.2. Cardiac MRI Parameters

Mean delay between cardiac MRI and RHC was 39 ± 82 days. A vortex was observed in all but 2 patients (92%) on 4D flow images (Figure 2). Patients with PH (defined as a mPAP > 20 mmHg on RHC) exhibited significantly higher right ventricular end systolic volumes, mean PA diameters and vortex durations than patients without PH. If mPAP ≥ 25 mmHg was used as a threshold to define PH, as recommended before 2019 [19], the same significant differences with the addition of right ventricular end diastolic volume and distensibility were noticed between patients with and without PH. All data are reported in Table 2 and Table 3.

### 3.3. Receiver Operating Characteristic Curve Analysis

To evaluate the performance of cardiac MRI in detecting PH, patients were subdivided in a positive group (with PH) and a negative group (without PH). The highest areas under the curve (AUC) for participants with PH as opposed to participants without PH were obtained for vortex duration measured with MRI with AUCs of 0.860 [95% CI: 0.637, 1] and 0.896 [95% CI: 0.597, 1] for a PH definition threshold of >20 mmHg and ≥25 mmHg, respectively. Right ventricular volumes; RVEF, RV cardiac index, mean diameter of PA and pulmonary distensibility showed lower AUCs but no significant differences (*p* > 0.1) were noted between AUCs (Table 4 and Table 5, and Figure 3). Youden index showed that a vortex duration of 8.6% of cardiac cycle provided a 95% sensitivity and an 83% specificity to detect PH (defined as a pulmonary pressure > 20 mmHg). If a cut-off of ≥25 mmHg was used to define PH, the vortex duration increased to 10.0% of cardiac cycle, providing a 100% sensitivity and a 78% specificity to detect PH.

### 3.4. Relationships between Vortex Duration, mPAP and Other RHC and MRI Parameters

In patients with pulmonary hypertension defined as mPAP > 20 mmHg (*n* = 19), vortex duration showed good correlation with the mPAP (r = 0.745; *p* < 0.0001) (Figure 4A). Linear regression analysis between vortex duration and mPAP revealed that the corresponding linear regression equation was mPAP (mmHg) = 19.2 + 0.7 × vortex duration (%). The comparison of the mPAP as measured by RHC and as calculated resulted in a nonsignificant bias of 0.18 mmHg. The SD of differences was 9.4 mmHg, and 95% limits of agreement were 18.0 and −17.4 mmHg (Figure 4B).

A weaker correlation between mean vortex duration and other RHC and MR parameters was also observed: PVR (r = 0.518; *p* = 0.002), PCWP (r = 0.556; *p* = 0.007), RV EDV (r = 0.467; *p* = 0.02), RV ESV (r = 0.551; *p* = 0.004), RVEF (r = −0.610; *p* = 0.001), mean diameter of PA (r = 0.470; *p* = 0.02) and PA distensibility (r = −0.360; *p* = 0.04).

Several MR parameters also displayed a correlation with the mPAP but in a lesser manner than vortex duration: RV ESV (r = 0.492; *p* = 0.01), RVEF (r = −0.695; *p* < 0.001), mean diameter of pulmonary artery (r = 0.485; *p* = 0.01) and pulmonary distensibility (r = −0.365; *p* = 0.07).

Multiple regression analysis showed that other MR parameters registered did not correlate independently with vortex duration and could not be associated with a multiparametric model to improve diagnosis performances.

### 3.5. Reproducibility

Reliability for the measurements of the vortex duration was found to be excellent with intra-observer ICC = 0.823 (95% confidence interval: 0.754–0.952) and inter-observer ICC = 0.788 (95% confidence interval: 0.733–0.942). 

## 4. Discussion

In this study, we report that patients with PH showed a vortical flow within PA on 4D MRI and that the vortex duration correlated with the mPAP as measured during RHC, suggesting that the mPAP could be non-invasively estimated by cardiac MRI. Our results confirm the potential of measuring vortex durations in patients with suspected PH in comparison to invasive measurements through RHC.

The vortex duration method was first introduced by Reiter et al. who went from using a single slice with 3D velocity encoding initially [16], in order to harvest signal-to-noise-ratio by using the in-flow effect. A 4D-flow approach was used more recently by this author [17,18] and several other groups [10,15], but this technique is not currently used in clinical practice. This present work is therefore important to further validate and promote such an approach. Having an automatic measurement of the vortex duration, as recently proposed [21,22], could also facilitate the development of this technique in clinical routine.

Recently, other approaches have been tested to assess PH from 4D-MRI. These schemes are looking for a relationship between PH and abnormalities on 4D flow images, but used more complex quantitative parameters (like vorticity or helicity) instead of visual analysis of vortex duration [8,23,24], making their use in clinical routine more difficult. These more complex approaches remain to be compared with the visual assessment of vortex duration in terms of accuracy and clinical feasibility. Interestingly, despite a visual analysis to measure vortex duration, which may reduce reproducibility, we reported a good inter-observer and intra-observer ICC.

The correlation between vortex duration and mPAP which we report was not as high as reported in two different series of patients with PH (defined as pulmonary pressure ≥ 25 mmHg), with correlation coefficients of 0.94 (AUC = 1) and 0.95 (AUC = 0.994) [16,17]. The potential explanations mainly include the different patient population and the timing between RHC and MRI. Indeed, most of our patients had a PH attributable to CTEPH unlike several studies in which this pathology was not the primary cause of PH [10,17,25]. Kamada et al. published a study including only patients with CTEPH and reported the interest of 4D flow MRI in such patients before and after treatment [8] but visual analysis of vortex duration was not performed, precluding direct comparison with our results.

The study of the ROC curves shows a higher AUC for the duration of the vortex. The lack of statistical difference with the other parameters is probably due to the relatively small number of patients. Interestingly our cut-off value for vortex duration to detect a PH with mPAP ≥ 25 mmHg was 10% of the cardiac cycle (sensitivity 100%, specificity 78%), in the same range (14.3%) as previously reported [17]. This value dropped to 8.6% using the newly suggested threshold of mPAP > 20 mmHg to diagnose PH [1], with values of sensitivity and specificity remaining high (95% and 83%, respectively). These values seemed to be at least equivalent or even superior to transthoracic echocardiography. A meta-analysis of the performance of transthoracic echocardiography showed a sensitivity of 88% and a specificity of 56% for the diagnosis of PH using the estimation of the velocity of tricuspid flow regurgitation [26]. However, the evaluation of the velocity of tricuspid flow regurgitation was not possible in 54 to 90% of the patients depending on the aetiology of PH and the morphology of the patients (e.g., especially obese patients) [26]. MRI could therefore be a valuable non-invasive tool for the screening of PH, especially in situations where transthoracic echocardiography is not able to estimate systolic PAP. Transthoracic echocardiography currently remains the best screening and follow-up tool for PH patients [27].

Several explanations have been advanced to explain the formation of vortices in PH, including reduction of arterial compliance and wall shear stress, increased PVR and endothelial cell dysfunction [28,29]. These abnormalities affect the laminar flow pattern of the PA causing turbulence and the eventual vortex. We report here a significant correlation between vortex duration, pulmonary distensibility and PVR that are consistent with these hypotheses. Similarly, Kheyfets et al. reported a significant correlation between vorticity in the main PA and PVR and proposed a multivariate linear regression model based on MRI parameters to estimate PVR [23]. Computational fluid dynamics has also suggested that a change from narrow to wider vessels (i.e., PA dilatation) could cause a vortex [30]. This could also explain the moderate correlation of mPAP and vortex duration with PA diameter which we observed. In addition, it has been reported that vortices can also be encountered in a low percentage (3%) of subjects without history of PH [31]. The relationship between right ventricular dysfunction and vortex duration in our study was also found by Odagiri et al. [10], and probably reflects the consequences of PH on the right ventricle. We did not notice significant differences between patients with and without PH regarding cardiac index, possibly because pulmonary dilatation in patients with PH could act as an adaptive phenomenon in order to maintain pulmonary flow, as reported in animals [10,28].

Our study has some limitations. First, the scan time for the 4D Flow sequence was relatively long. However, acceleration techniques, such as compressed sensing are emerging and will allow such sequences to fit more easily into the clinical routine [32,33]. Second, the encoding velocity used for the 4D flow (150 cm/s) was significantly higher than the average speed of the patients (63 cm/s), which may have led to poorer visualisation of vortices. As vortices have a large range of velocities, multi-venc acquisition techniques in combination with acceleration techniques may be of interest for the proposed application [34]. Lastly, RHC and cardiac MRI were not performed on the same day in most of the patients, and it is possible that variations of mPAP could have occurred between the two examinations.

In conclusion, our results show a relationship between vortex duration in the pulmonary artery and pulmonary arterial pressure, suggesting that this non-invasive parameter could be used to estimate pulmonary arterial pressure.

## Figures and Tables

**Figure 1 jcm-11-05237-f001:**
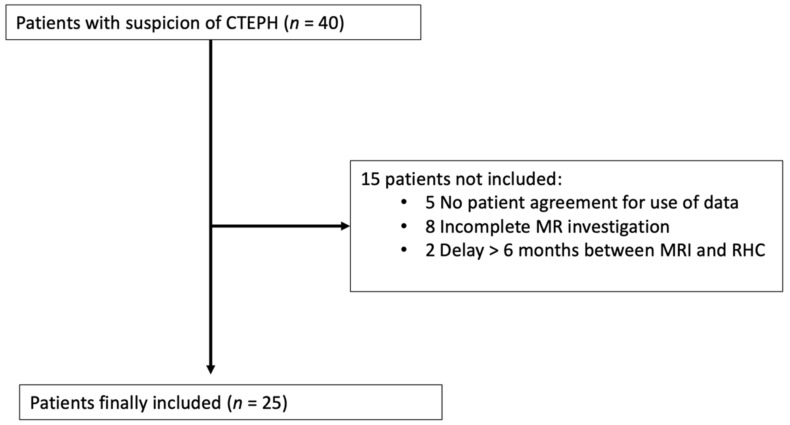
Flow chart of the study.

**Figure 2 jcm-11-05237-f002:**
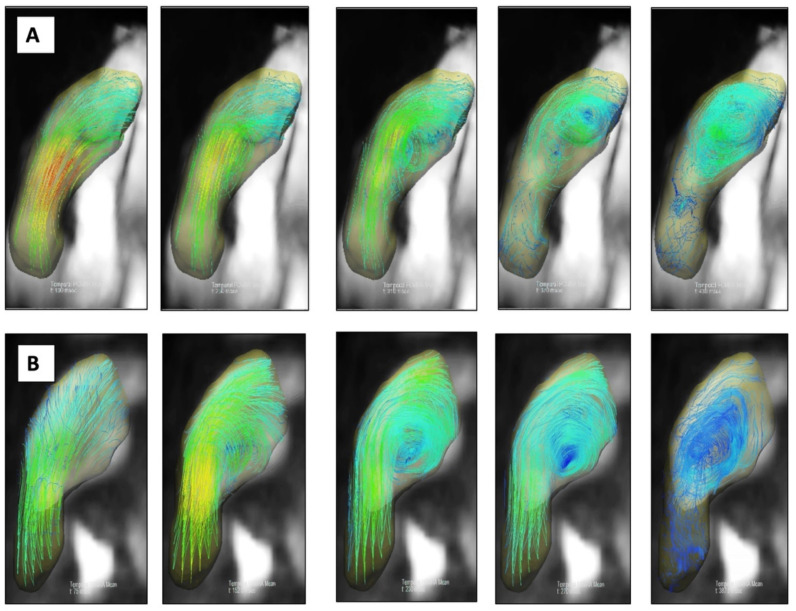
Examples of two patients with CTEPH, PH and vortical flow on 4D flow MR images. Patient 1 ((**A**) first image row) exhibited a vortical flow within pulmonary artery, detected during 60% of the cardiac cycle. Mean PAP was 46 mmHg by RHC. Patient 2 ((**B**) second image row) exhibited a vortical flow within pulmonary artery, detected during 32% of the cardiac cycle. Mean PAP was 26 mmHg by RHC.

**Figure 3 jcm-11-05237-f003:**
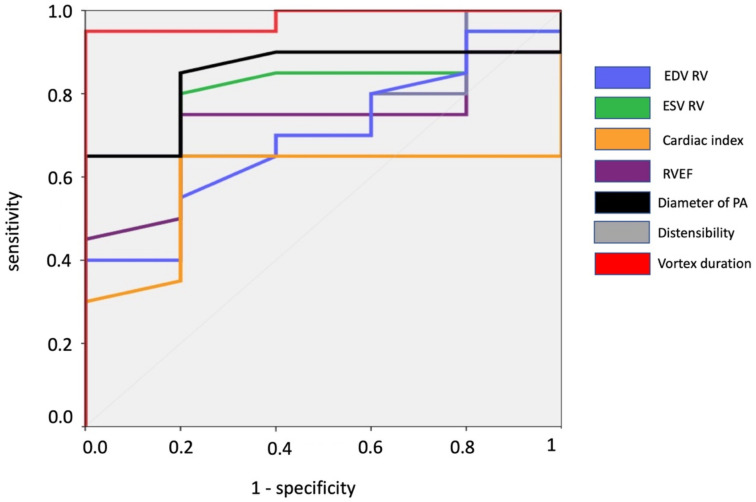
ROC curves analysis to detect PH in patients. The cut-off for definition of PH was 20 mmHg.

**Figure 4 jcm-11-05237-f004:**
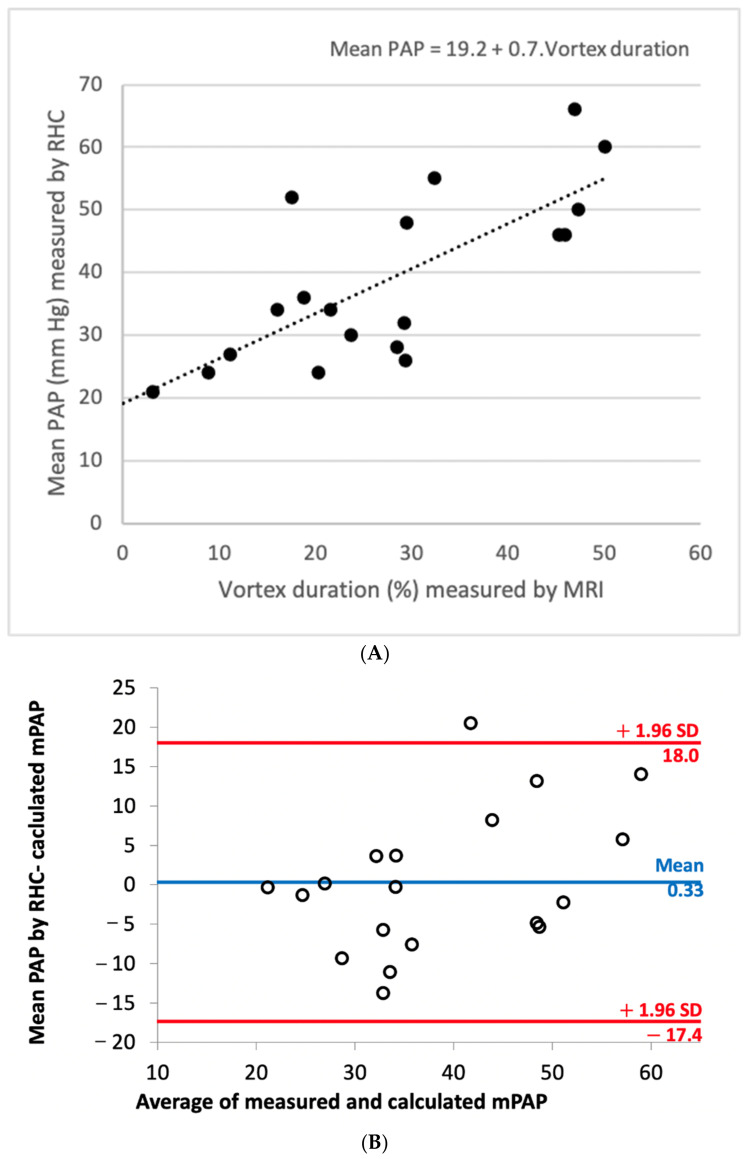
(**A**) Correlation and linear regression line between vortex duration (%) measured by MRI and mPAP (mmHg) measured by RHC. (**B**) Bland–Altman plot of mPAP measured by RHC and calculated by the regression equation mPAP (mmHg) =19.2 + 0.7 × vortex duration (%), with 95% limits of agreement.

**Table 1 jcm-11-05237-t001:** Population characteristics.

A. Clinical Parameters	
Age, years	63 ± 16
Men, *n* (%)	10 (40)
Overweight, *n* (%)	5 (20)
HBP, *n* (%)	7 (28)
Diabetes, *n* (%)	4 (16)
WHO functional class, *n* (IQR)	2 (2–3)
**B. Right Heart Catheterization**	
Heart rythm, bpm	83 ± 13
Mean PAP, mm Hg	33 ± 15
PCWP, mm Hg	7.4 ± 3.9
LVEDP, mm Hg	10.2 ± 4.7
PVR, Wood unit	5.8 ± 3.9
Cardiac output, L/min	4.4 ± 1.2
Cardiac index, L/min/m^2^	2.4 ± 0.8

HBP = high blood pressure, LVEDP = left ventricle end diastolic pressure, PCWP = pulmonary capillary wedge pressure, PVR = pulmonary vascular resistance.

**Table 2 jcm-11-05237-t002:** Cardiac MRI parameters in the overall population and between patients with and without pulmonary hypertension on RHC (defined as a mean pulmonary pressure > 20 mmHg).

	All (*n* = 25)	PH (*n* = 19)	No PH (*n* = 6)	*p*
Mean heart rate, bpm	79 ± 17	82 ± 18	72 ± 8	0.08
RV EDV, mL/m^2^	95 ± 43	101 ± 48	75 ± 14	0.1
RV EDS, mL/m^2^	52 ± 35	58 ± 38	34 ± 10	0.02
RV EF, %	48 ± 12	46 ± 13	55 ± 6	0.2
RV cardiac index, L/min/m^2^	2.9 ± 0.2	3.0 ± 0.9	2.5 ± 0.6	0.2
Mean PA diameter, mm	32 ± 6	33 ± 7	29 ± 2	0.01
PA distensibility, %	8.6 ± 6.6	7.3 ± 5.3	12.9 ± 9.1	0.07
Mean peak velocity in PA, cm/s	63 ± 20	61 ± 23	70 ± 7	0.1
Vortex duration, %	23 ± 16	27 ± 14	8.3 ± 13	0.007

EDV = end diastolic volume; EF = ejection fraction; ESV = end systolic volume; PA = pulmonary artery; PH = pulmonary hypertension; RV = right ventricle.

**Table 3 jcm-11-05237-t003:** Cardiac MRI parameters in the overall population and between patients with and without pulmonary hypertension on RHC (defined as a mean pulmonary pressure ≥ 25 mmHg.)

	All (*n* = 25)	PH (*n* = 16)	No PH (*n* = 9)	*p*
Mean heart rate, bpm	79 ± 17	84 ± 19	73 ± 6	0.07
RV EDV, mL/m^2^	95 ± 43	107 ± 50	73 ± 16	0.03
RV EDS, mL/m^2^	52 ± 35	63 ± 40	33 ± 11	0.004
RV EF, %	48 ± 12	44 ± 13	56 ± 7	0.03
RV cardiac index, L/min/m^2^	2.9 ± 0.2	3.0 ± 0.9	2.7 ± 0.6	0.6
Mean PA diameter, mm	32 ± 6	35 ± 4	28 ± 7	0.008
PA distensibility, %	8.6 ± 6.6	6.6 ± 5.3	12.2 ± 7.5	0.02
Mean peak velocity in PA, cm/s	63 ± 20	61 ± 24	63 ± 20	0.2
Vortex duration, %	23 ± 16	30 ± 13	9 ± 11	0.001

EDV = end diastolic volume; EF = ejection fraction; ESV = end systolic volume; PA = pulmonary artery; PH = pulmonary hypertension; RV = right ventricle.

**Table 4 jcm-11-05237-t004:** Receiver operating characteristic curve analysis of MR parameters to detect pulmonary hypertension (mean pulmonary pressure > 20 mmHg on RHC).

	AUC	95% CI	95% CI
RV EDV (mL/m^2^)	0.715	0.509	0.921
RV ESV (mL/m^2^)	0.803	0.619	0.988
RV EF (%)	0.697	0.495	0.900
RV cardiac index (L/min/m^2^)	0.680	0.455	0.905
Mean diameter of PA (mm)	0.842	0.687	0.998
PA distensibility (%)	0.754	0.563	0.946
Vortex duration	0.860	0.637	1

AUC = area under curve; RV = right ventricle; EF = ejection fraction; EDS = end diastolic volume; ESV = end systolic volume; PA = pulmonary artery.

**Table 5 jcm-11-05237-t005:** Receiver operating characteristic curve analysis of MR parameters to detect pulmonary hypertension (defined a mean PAP ≥ 25 mmHg).

	AUC	95% CI	95% CI
RV EDV (mL/m^2^)	0.767	0.582	0.952
RV ESV (mL/m^2^)	0.844	0.690	0.998
RV EF (%)	0.760	0.571	0.995
RV cardiac index (L/min/m^2^)	0.563	0.331	0.794
Mean diameter of PA (mm)	0.819	0.643	0.996
PA distensibility (%)	0.785	0.403	0.972
Vortex duration	0.896	0.597	1

AUC = area under curve; RV = right ventricle; EF = ejection fraction; EDS = end diastolic volume; ESV = end systolic volume; PA = pulmonary artery.

## Data Availability

The datasets used during the current study are available from the corresponding author on reasonable request.

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
