# Peer review of "Correlation between Pulmonary Artery Pressure and Vortex Duration Determined by 4D Flow MRI in Main Pulmonary Artery in Patients with Suspicion of Chronic Thromboembolic Pulmonary Hypertension (CTEPH)"

_jcm, 2022, doi:10.3390/jcm11175237_

Round 1

Reviewer 1 Report

General Comments:

Authors investigated 25 patients with suspected CTEPH using RHC and comprehensive cardiac MRI to evaluate the applicability of cardiac MRI parameters for the diagnosis of PH (mPAP>20 mmHg as well as mPAP³25 mmHg as thresholds) and the non-invasive estimation of mPAP. Cardiac MRI included cine function, 2D main pulmonary artery through-plane flow and 4D-flow imaging. Authors report a strong correlation between RHC-derived mPAP and duration of vortical blood flow in the main pulmonary artery in CTEPH patients similar to results found previously in PH-populations including all PH groups. This result is of high clinical interest, there are, however, some general aspects which should be addressed/improved:

1.       25 subjects might be not enough to compare different CMR parameters for diagnosis of PH. The confidence intervals of AUCs are large and overlap, indicating no significant difference between parameters. Stating that 4D-Flow PH-vortex-based diagnosis of CTEPH and mPAP-estimation is feasible can be deduced from the results (Discussion L259). Stating that among different MR parameters to estimate mPAP 4D-Flow was the most relevant is misleading (Conclusion L343) as significance was not reached in this (small) study population. Please address this aspect.

2.       Inclusion/Exclusion of subjects: How many patients with suspicion of CTEPH were investigated by CMR and RHC (of which 25 were analyzed)? Was there any change in treatment or patient state between RHC and CMR (within the 6 months between investigations)? Where patients with irregular heartbeat (RR-interval) included? If these subjects were included, was there any difference in the cardiac MR imaging protocols?

3.       Various cardiac MR parameters have been evaluated for non-invasive diagnosis of PH (for mPAP >20 mmHg and mPAP³25 mmHg); mPAP estimation, however, was derived from 4D-flow vortex duration only. Please rephrase the Conclusion accordingly.

4.       The application of a linear regression model for the relationship of mPAP on vortex duration is dubious, because it assumes that even for small/normal mPAPs there is a vortex duration > 0. It would be more correct to either restrict the model to CTEPH patients or use a piece-wise linear fit of vortex duration on mPAP.

Specific comments:

 Abstract

- L37: Youden test should mean Youden index?

 Introduction

- L56 and L58: “RHC remains the gold standard …” is true but repetitive. Please rephrase.

- L65/66: Please rephrase “duration … were”: either “duration … was” or “durations … were”

- L68: The term vorticity is incorrect as it represents a fluid-mechanical metrics which was not investigated/calculated in the present study. Please rephrase e.g. to “duration of vortical blood flow in the main pulmonary artery”

Materials

- L75: “consecutive patients”: how many patients with suspected CTEPH were investigated? (see also general comments)

- L79: You term the study “a retrospective study”.  Was 4D-flow assessed in subjects routinely? Please comment.

- L80: It would be important to state if patients with changes in therapy or changes in disease-condition within the 6 months where excluded; or if there were no changes in therapy and/or conditions between RHC and CMR. Were patients with irregular RR-interval included? If so, please state. (see also general comments)

- L85: was the cine a “balanced steady state free precession (bSSFP)? If so, please correct.

- L89 and L96: I don’t really understand why the temporal resolution of cine and 4D-Flow was different according cardiac rhythm. Please clarify.

- L93: A VENC of 150 cm/s for CTEPH appears very high. What was the peak velocity in the main pulmonary artery? It would be interesting to add peak velocities to the result Tables 2 and 3.

- L98: Similar to 2D-Flow, a VENC or 150 cm/s seems very high for CTEPH. For low velocities in the main pulmonary artery (as typically observed in CTEPH with high mPAP) such VENC could crucially impact the visibility of vortical blood flow patterns. If VENC was high compared to peak velocities in the main pulmonary artery, this should be discussed and/or added to the limitation section.

- L123: The term “pulmonary artery surfaces” is incorrect, it should be “pulmonary artery cross-section area”

- L133: Please indicated if you used background phase correction.

- L157: “Youden test” sounds strange and should rather be the “Youden index”.

- P4, Table 1: Echo-Data, 6 min walk test and blood test data are included. This is missing in the methods (together with time between other investigations). As they are not further investigated you could also delete these results in Table 1 or add these tests to the methods, results and discussion sections. Please add also the average heart rate or RR interval during RHC to Table 1.

- P6, Table 2: Please add average heart rate or RR interval during CMR to Table 2.

- L201ff: AUC analysis with such small number of subjects is difficult to interpret (see also general comments). CI in Table 4 and Table 5 overlap, indicating that AUCs not differ significantly. This needs to be discussed/added to the limitations section.

- L252f: I don’t understand the statement: “No significant difference (P > 0.7) was noticed between measurements which were significantly correlated with each other (R2>0.8; P<0.0005).” Please rephrase.

Discussion

The Discussion section could benefit from “streamlining” of topics, as various aspects appear “repetitively” (sequence technique, other groups, …). Moreover the discussion of non-vortex parameters for detection of PH and mPAP-estimation is lacking (volumetric function, 2D-Flow) and could be discussed with respect to literature.

- L261: There is at least the group of Martin Ugander/SWE, Jan Kröger/DE, Hideki Ota/JAP, Yuchi Han/USA, and some more. Please rephrase statement.

- L263: In fact Ref 15-17 use the same sequence technique. Please rephrase statement.

- L271: Please rephrase as CTEPH subjects were included in studies [16,17,14,29] (even though in smaller numbers than PAH).

- L277: Take maybe into account Ramos et al. BMC Medical Imaging 2020.

- L303: See Ramos et al. BMC Med Imaging

- L330ff: This is paragraph is not a limitation (as metrics were not evaluated) and should be moved to the discussion.

- L228: VENC could represent a limitation (if it was high compared to maximal velocity observed in the study population). Please check.

Conclusion:

See general comments

Reviewer 2 Report

This retrospective study on 25 patients with suspected CTEPH shows a good correlation between Tvortex on 4D flow MR sequences and mPAP measured on RHC. This is true for both the new and the old definition of PH. This type of study was long expected since the results of Reiter et al. in 2008 and 2015 showing a correlation of 0.94 and 0.95 between tvortex and mPAP, and from the same author.

The major pitfalls of this study were the delay between CMR and RHC 39 +/- 82 days, the small number of patients included, and the lack of an external validation data set.

Introduction : concise.

Methods :

P3L136 Briefly provide more details on the methology for calculating the vortex duration. When does the measurement start, when does it end.

p4L160 Please define ICC literally in the statistical analysis and add this citation : 

Benchoufi, M., Matzner-Lober, E., Molinari, N., Jannot, A. S., & Soyer, P. (2020). Interobserver agreement issues in radiology. Diagnostic and interventional imaging101(10), 639-641.

Results :

P4L163 Please provide a flow chart showing the number of patients excluded (due to incomplete CMR or delay > 6 month between CMR and RHC). It could be difficult to stay more than 30 minutes in the magnet, breathing quietly with occasional apnea, in a patient with WHO functional class 2-3. We can assume that your patients are highly selected from the suspected CTEPH patients.

P4L170 Please provide the diagnosis for the 6 patients without CTEPH.

Medication use could be a confounding parameter. For example, if the patient starts medical therapy within the time between the RHC and the CMR. Please provide the time between RHC and the first intake of medical therapy.

P9L252 Please provide the 95% CI for ICC inter and intra observer.

Discussion: Very interesting and precise. I fully agree with authors p10L270 to 273.

Table and figures : OK
